# Darwin Returns to the Galapagos: Genetic and Morphological Analyses Confirm the Presence of *Tramea darwini* at the Archipelago (Odonata, Libellulidae)

**DOI:** 10.3390/insects12010021

**Published:** 2020-12-31

**Authors:** María Olalla Lorenzo-Carballa, Rosser W. Garrison, Andrea C. Encalada, Adolfo Cordero-Rivera

**Affiliations:** 1ECOEVO Lab, E.E. Forestal, Campus Universitario A Xunqueira s/n, University of Vigo, 36005 Pontevedra, Spain; malorenzo@uvigo.gal; 2California Department of Food & Agriculture, 3294 Meadowview Road, Sacramento, CA 95832-1448, USA; argiavivida@gmail.com; 3Laboratorio de Ecología Acuática, Instituto BIOSFERA, Diego de Robles y Vía Interoceánica, Campus Cumbayá, Universidad San Francisco de Quito, 1712841 Quito, Ecuador; aencalada@usfq.edu.ec

**Keywords:** dragonflies, taxonomy, islands, molecular markers, morphological analysis, synonymy

## Abstract

**Simple Summary:**

Flying insects are able to colonize oceanic islands by both active and passive dispersal. Ten species of dragonflies are found in the Galapagos archipelago, located at 900 km from mainland South America. Shortly after the publication of Darwin’s “The Origin of Species”, one of the dragonflies from these islands was named after him as *Tramea darwini*. However, subsequent studies considered it to belong to another continental species of the same genus known as *Tramea cophysa*. Here, we studied a series of specimens of *Tramea* collected in 2018 from the Islands of San Cristobal, Isabela, and Santa Cruz, with the aim of determining their specific identity, through a combination of molecular and morphological analyses. Our results indicate that the Galapagos specimens examined belong to *Tramea calverti*, another continental species, and not to *T. cophysa* as previously thought. Following the principle of priority in taxonomic nomenclature, *Tramea calverti*, which was described in 1910 by Muttkowski, should hereafter be considered a synonym of *Tramea darwini*, which was described in 1889 by Kirby; hence, the species named after Darwin is to be considered valid, inhabiting both the Galapagos islands and continental America.

**Abstract:**

The status of the *Tramea* species present in the Galapagos Islands (Odonata, Libellulidae) has been the subject of a long-standing debate among odonatologists. Here, we use molecular and morphological data to analyze a series of specimens from this genus collected in 2018 from the Islands of San Cristobal, Isabela, and Santa Cruz, with the aim of determining their relationship with *Tramea calverti* Muttkowski and with their currently considered senior synonym *T. cophysa* Hagen. We combined sequencing of mitochondrial and nuclear DNA with morphological examination of several specimens of *Tramea*, including representatives of continental *T. cophysa* and *T. calverti*. Our molecular analyses place the *Tramea* from Galapagos in the same clade as *T. calverti*, with *T. cophysa* as a closely related species. The morphological analyses found only one consistent difference between *T. cophysa* and *T. calverti*: the presence of an accessory lobe in the male vesica spermalis of *T. cophysa* that is absent in *T. calverti* and in the *Tramea* from Galapagos. In agreement with our genetic results, the overall morphological differences documented by us indicate that the Galapagos material examined is conspecific with *T. calverti*. In light of this, and following the principle of priority in taxonomic nomenclature, *Tramea calverti* Muttkowski, 1910 should hereafter be considered a junior synonym of *Tramea darwini* Kirby, 1889.

## 1. Introduction

Islands are laboratories of evolution in action, mainly due to their isolation and small populations, which drive rapid changes in the biota [1]. For these reasons, islands—especially oceanic islands never connected to continents—have attracted the interest of many early naturalists. Already in 1855, de Candolle described the fact that islands have a poorer biota when compared to similar-sized areas of the continents [2], which provided the basis for current theories about island biogeography [3]. In his famous voyage on the Beagle, Darwin visited the Galapagos, where he was impacted by the biota and geology of these islands, which greatly influenced his thoughts about natural selection and evolution [4].

Since the second half of the 19th century, the Galapagos flora and fauna became the subject of more than 20 expeditions, which highlighted the fact that the insect fauna of this archipelago is poor, as expected given its geographical isolation [5]. In the case of the insect order Odonata (dragonflies and damselflies), the diversity is greatly reduced, with only 10 species recorded so far [6,7,8]. The only odonate species endemic to the Galapagos is *Rhionaeschna galapagoensis* (Currie), a member of the family Aeshnidae [9]; and the only zygopteran species found in the archipelago is *Ischnura hastata* (Say) [10]. Seven of the remaining species belong to the family Libellulidae, which includes some species of dragonflies with a wide distribution, like *Pantala flavescens* (Fabricius), a cosmopolitan species that occurs on all continents (except Antarctica and most of Europe).

Additionally within the Libellulidae, the genus *Tramea* has been reported for the Galapagos archipelago. This genus is, along with *Pantala* and some members of the family Aeshnidae, a well-known migrant. The characteristic broad based hindwings present in the species of this genus, allow them to perform long-distance flights at a reduced energetic cost and, in fact, several instances of large swarm migrations have been reported for *Tramea* species in tropical regions [11]. The genus *Tramea* is worldwide distributed. In the American continent, *Tramea* species are found from Canada to Uruguay and central Argentina [12].

Two species of *Tramea* have been listed for the Galapagos islands: *Tramea calverti* Muttkowski, 1910 and *T. darwini* Kirby, 1889 [7,13]. However, the most recent and exhaustive revision of the *cophysa* group carried out by De Marmels and Rácenis [14], lists only *T. cophysa* from the archipelago, after the authors examined two females of *T. darwini* collected at the islands of Isabela and Floreana.

Ever since its original description based on five females (only one in good condition) deposited in the British Museum, the taxonomic status of *Tramea darwini* [15] has been discussed by odonatologists. Both Calvert [16] and Ris [17] considered this species a synonym of *T. cophysa* Hagen, 1867, and this practice has been followed until today [14]. However, and despite a number of past publications dealing with the *Tramea* species from the Galapagos [16,17,18,19,20,21,22] there has been no consensus as to what name should be given to these insular populations. Part of the problem may lie in the lack of a sufficient series of insular material with which to compare with either of the mainland taxa.

Our purpose here is to determine the relationships among the taxa currently going under the names *T. calverti*, *T. cophysa,* and *T. darwini*, based on a sufficient series of specimens from the latter collected on the Galapagos in 2018. To do this, we sequenced nuclear and mitochondrial DNA from representatives of the three taxa and carried out a morphological analysis, comparing individuals of *T. darwini* from Galapagos with *T. cophysa* and *T. calverti* collected at several locations within their continental distribution ranges.

## 2. Materials and Methods

### 2.1. Specimen Collection, DNA Extraction, and Sequencing

A total of 19 specimens of *T. darwini* were collected during a field trip to the Galapagos Islands of Isabela, Santa Cruz, and San Cristobal in August 2018 by AC-R and MOL-C (see Appendix A for details on collection locations). Adult individuals were captured with a hand net and placed in 80% ethanol until DNA extraction. Legs from dried specimens of several *Tramea* species (*T. cophysa*, *T. calverti, Tramea abdominalis* (Rambur, 1842)*, Tramea basilaris* (Palisot de Beauvois, 1817), *Tramea binotata* (Rambur, 1842)*, Tramea carolina* (Linnaeus, 1763)*, Tramea lacerata* Hagen, 1861 and *Tramea virginia* (Rambur, 1842)) belonging to RWG’s personal collection were also used for DNA extraction (see Appendix A).

Total genomic DNA was extracted from individual legs using the GeneJet DNA extraction kit (ThermoFisher Scientific, Waltham, MA, USA), following the manufacturer’s protocol. To reconstruct the phylogenetic relationships between the *Tramea* species, we amplified fragments of the mitochondrial *16S* and Cytochrome Oxidase I (*COI*) genes, together with the nuclear Internal Transcribed Spacer (*ITS*), using previously published primers [23,24,25] (see Appendix A). PCR reactions were carried out using the DreamTaq Green PCR Master Mix (ThermoFisher Scientific, Waltham, MA, USA). Prior to sequencing, unincorporated primers and dNTPs were removed using Shrimp Alkaline Phosphatase and Exonuclease I (New England Biolabs, Ipswich, MA, USA). Cleaned PCR products were sequenced in both directions using BigDye v.3.1 chemistry (Applied Biosystems, Foster City, CA, USA) in an ABI 3730xl DNA Analyzer (Applied Biosystems, Foster City, CA, USA), by the Macrogen Spain sequencing services.

### 2.2. Genetic Analyses

After sequencing, chromatograms were visually inspected, trimmed and automatically assembled using Geneious v. 9.1.7 (https://www.geneious.com). For some of the ITS sequences, we obtained superimposed traces, characteristic of sequences containing heterozygous insertions/deletions (indels). Allelic sequences were reconstructed using Indelligent v.1.2 [26]. We run BLAST searches for all DNA sequences at the NCBI website (https://blast.ncbi.nlm.nih.gov/Blast.cgi), to ensure that they were not derived from contaminations.

Sequences were aligned using MAFFT [27,28], as implemented in Geneious v 9.1.7. Genetic differentiation between species (p-distance) was estimated for each dataset in MEGA X [29], using the pairwise deletion option, which removes all ambiguous positions for each sequence pair. Mitochondrial DNA (mtDNA) alignments were concatenated for phylogenetic analyses. Phylogenetic relationships among *Tramea* species were reconstructed using maximum likelihood (ML) and Bayesian inference (BI) approaches. To increase the robustness of the analyses, previously published sequences from several *Tramea* species downloaded from GenBank (https://www.ncbi.nlm.nih.gov/genbank/) were added to our datasets. *Pantala flavescens* was selected as the outgroup for the phylogenetic analyses (see Appendix A).

ML analyses were carried out using RAxML 7.2.8 [30,31] as implemented in Geneious v 9.1.7., using the rapid bootstrapping and search for best scoring ML tree option, under the GTR + I + G model. Support for the nodes was estimated by running 1000 bootstrap replicates. For BI analyses, we used MrBayes 3.2.6 [32,33], as implemented in Geneious v 9.1.7. BI searches were run for 1.1 million generations, with default priors and with the GTR + I + G substitution model. Resulting phylogenetic trees were edited with TreeGraph 2 [34].

To further confirm the species delimitation within our datasets, we used the single locus distance-based delimitation methods implemented by Automated Barcode Gap Detection (ABGD) [35]. Analyses were run at the ABGD web server (https://bioinfo.mnhn.fr/abi/public/abgd/abgdweb.html). Fasta files including the aligned ingroup sequences (i.e., the sequences from the *Tramea* species) for each locus (*ITS*, *COI,* and *16S*) were used as input files for the analyses, which were carried out with the default options.

### 2.3. Morphological Analyses

We examined ten individuals of *Tramea* from the Galapagos, to determine if any morphological characters would corroborate the placement of these populations under *T. calverti* or *T. cophysa*: two individuals from Isabela island (1 ♂, 1 ♀), six individuals from San Cristobal island (3 ♂♂, 3 ♀♀) and two from Santa Cruz island (1 ♂, 1 ♀). Our material from Galapagos was compared to 22 individuals of *T. cophysa* (15 ♂♂, 7 ♀♀) from southeastern Brazil and northern Argentina, and 46 individuals of *T. calverti* (27 ♂♂, 19 ♀♀) ranging from Arizona (USA) south to northern Argentina. We also examined specimens of both species taken together at Ilha de Marambaia in Rio de Janeiro State, Brazil in order to further determine whether the morphology of the *Tramea* from Galapagos matched either *T. cophysa* or *T. calverti*.

Specimens of *T. cophysa* and *T. calverti* belonging to RWG’s collection were killed by injecting them with acetone, in order to preserve color patterns and to prevent lateral pressure distortions. Afterwards, they were placed in envelopes and steeped in acetone for 24 h, to promote drying. The *Tramea* specimens collected at the Galapagos by MOL-C and AC-R, which were preserved in 80% ethanol, were injected with acetone and steeped in acetone for 24 h, prior to morphological analyses.

Specimens were examined under a Zeiss Discovery V20 Stereo Microscope at magnifications ranging from 7.5× to 150×. Entire specimens and wings were scanned at 1200 DPI using an Epson Perfection V600 Photo Scanner. Heads and abdomens were photographed using a Leica MC 170 HD digital camera attached to the microscope at varying magnifications and stacked using Helicon Focus software. Vesicas were removed from each male, soaked in hot water, and cleaned with a brush so that the various lobes and details of this structure could be observed. Using a pair of watch forceps, resulting preparations were temporarily pinned with 0.10 minutens and fastened to small lumps of dentist’s wax submerged in 95% ethanol and illustrated via a camera lucida. 

The following morphological characters were examined: Head—coloration of vertex, postfrons and labrum in males and females; wings—coloration of wing membrane and extent of Hw basal spot in males and females; vesica spermalis; abdomen—coloration of ventral tergites; and S8 in males and females (abbreviations: Hw = hind wing; S = abdominal segment).

## 3. Results

### 3.1. Genetic Analyses

The BLAST searches identified all our obtained sequences as similar to other odonate sequences available in the NCBI database. The final datasets (excluding the outgroup, *P. flavescens*) consisted of 44 *COI* and *16S* sequences and 37 *ITS* sequences. The *16S* dataset was 505 bp long, with 39 variable sites and 32 parsimony informative sites; the *COI* dataset was 451 bp long, with 98 variable sites and 93 parsimony informative sites. The *ITS* dataset was 894 bp long (including gaps), with 188 variable sites and 85 parsimony informative sites. All the sequences generated in this study have been deposited in the GenBank sequence database (https://www.ncbi.nlm.nih.gov/genbank), under accession numbers MW246873-MW246955 (see Appendix A).

Results of phylogenetic analyses were congruent for both datasets and also for the BI and ML analyses, placing *T. darwini* in the same clade as the continental *T. calverti*, while *T. cophysa* appears as a sister/closely related species. These relationships were supported by high posterior probability and bootstrap values (Figure 1).

In agreement with the results of the phylogenetic analyses, the interspecific distances for both the mitochondrial and nuclear datasets group *T. darwini* with *T. calverti*. The average distances between *T. darwini* and *T. calverti* were 0% for *16S*, 0.4% for *COI* and 0.3% for *ITS*; while the average distances between *T. darwini* and *T. cophysa* were 1.3% for *16S*, 3.3% for *COI,* and 6.7% for *ITS* (see Appendix A).

The ABGD species delimitation analyses identified nine groups (i.e., phylogenetic species) for the mtDNA loci and eight groups for the nDNA locus (see Supplementary Information Data S1). The groups comprised by (i) *T. cophysa* and (ii) *T. darwini* and *T. calverti* were both recovered as different species in all cases (Figure 1).

### 3.2. Morphological Analyses

We found only one morphological difference between *T. cophysa* and *T. calverti*. The vesica spermalis of *T. cophysa* in lateral view possesses an accessory lateral lobe (Figure 2c,d), which is absent in *T. calverti* (Figure 2a). The vesica in specimens of *Tramea* from the Galapagos also lacks the accessory lateral lobe (Figure 2b). We found no consistent differences in the morphology of the male hamules and cerci nor in female morphologies.

The vertex, postfrons, and labrum consistently differed in coloration, only in the males, as follows: the dorsal surface of the vertex in *T. cophysa* is dark metallic violet matching the coloration of the postfrons (Figure 3a,b). The dorsal surface of the vertex in *T. calverti* is pale brown (may be obscured in postmortem preservation) and differs from the dark metallic violet luster of the postfrons (Figure 3g,h,j,). Additionally, the metallic violet luster in *T. cophysa* extends anteriorly covering the entire postfrons. In *T. calverti*, the dark metallic violet luster is confined to about the basal half of the postfrons. The labrum is mostly dark brown in males of *T. cophysa* (Figure 3b) but mostly pale brown in males of *T. calverti* (Figure 3h,j). Males of *Tramea* from the Galapagos consistently matched the coloration of *T. calverti* (Figure 3d,f). We detected no consistent difference in female head coloration between the two species (Figure 3c,e,i,k).

The wing membrane in *T. cophysa* is hyaline in both sexes, and the Hw basal spot is dark brown and can be variable in extent (see Figure 4a–d and Figure 5a,b). The Hw spot in females is often reduced occupying the basal half or less of the length of the Hw base (Figure 4b,d and Figure 5b). In mainland populations of *T. calverti* the wings are slightly infused with amber (Figure 4e–h,m,n and Figure 5d,e) and the Hw spot is of a slightly lighter brown and is more extensive in both sexes. The size and extent of the Hw patch in mainland populations tends to be uniform exhibiting less variability than in *T. cophysa* (Figure 4e–h,m,n and Figure 5d,e). The Galapagos populations exhibit characters of both species, with the Hw patch being variable as in *T. cophysa* and the wing membrane having less of an infusion of amber coloration compared to mainland populations (Figure 4i–l and Figure 5c,d).

Coloration of the ventral abdominal tergites generally differs between *T. cophysa* and *T. calverti*. The majority of the ventral tergites in both sexes of *T. cophysa* are dark brown and of the same color as the lateral carinae in most of the abdominal segments (Figure 6a,b). The ventral tergites of some females (Figure 6c) may possess a lighter color differing from the dark lateral carinae but in those females, a light dusting of pruinosity is usually present on the more anterior segments (Figure 6c). The ventral tergites in *T. calverti* are always pale brown, with dark markings confined to the vicinity of the lateral carinae (Figure 6d–g). No pruinosity was observed in any of the *T. calverti* examined. Abdominal S8 is entirely dark brown in males of *T. cophysa* (Figure 7a,b), and the same is often observed in the females of this species (Figure 7c); but, in some females S8 shows a pale coloration confined to the lower half of the segment (Figure 7d). Both sexes of *T. calverti* consistently have an inverted pale brown semicircular spot at the base of S8 (Figure 7e,f), which is red or brown in live specimens. Specimens of *Tramea* from the Galapagos (Figure 7g,h) were consistent in coloration and pattern with *T. calverti*.

## 4. Discussion

The results of our genetic analyses show that the samples of *T. darwini* collected by us at three islands in the Galapagos archipelago (San Cristobal, Isabela, and Santa Cruz), all belong to a clade that also includes the mainland species currently known as *T. calverti*, with *T. cophysa* as a closely related species. The phylogenetic relationships are concordant between nuclear and mitochondrial DNA and supported by high bootstrap and posterior probability values in both cases. Furthermore, the genetic distances between the *Tramea* from Galapagos and *T. calverti* are also lower than the interspecific distances between the Galapagos individuals and *T. cophysa* (see Results and Appendix A). If we consider the 2% threshold commonly used as a limit between different species [36], we can conclude that the *Tramea* from Galapagos belong to *T. calverti* and not to *T. cophysa*. ABGD species delimitation methods provided further support for the placement *of T. darwini* and *T. calverti* within the same group, separated from *T. cophysa*.

In agreement with the molecular results, our morphological analyses indicate also a closer relationship between the material collected in the Galapagos islands and mainland *T. calverti*. This includes also the single morphological character noted above that consistently differentiates both *T. calverti* and *T. cophysa*: the accessory lobe that occurs in the male sperm vesica of *T. cophysa* (Figure 2c,d) but is absent in both *T. calverti* (Figure 2a) and the *Tramea* from Galapagos (Figure 2b).

In their exhaustive work on the *T. cophysa* complex, De Marmels and Rácenis [14] treat *T. darwini* Kirby, 1889, as a junior synonym of the older name, *T. cophysa* Hagen, 1867; following Calvert [16] (p. 303) who stated: “*Tramea darwini* based on a female with a much reduced basal wing marking has been shown by Mr. Currie [1901: 386] to vary greatly in this respect. His material is before me and I cannot separate it from some of the examples from Ecuador, while the Haytian female captured by W. Cabot has the basal brown spot of the hind wings reaching no farther backward than 1 mm. beyond the apex of the membranule.” De Marmels and Rácenis had access to only two females of *T. darwini* (Galapagos Is.: Albemarle, 6.VIII.1955; and Charles), a fact that may have accounted for their placement of the Galapagos specimens under *T. cophysa*.

The original description of *T. cophysa* states: “Die drei letzten Ringe oben und die vorhergehenden längs der Bauchkante schwarz; unten vom dritten Ringe an schwarz [Body brown, the last three abdominal segments at the top and those entirely along the edge of the abdomen black; at the bottom black from the third abdominal segment.]”, a characteristic not present in *T. calverti*.

A clue to the true identity of the Galapagos material was partially rectified by Peck [13] (pp. 313 and 316) as follows: “[*T. cophysa*] may have been confused with *T. calverti*, and its literature records may refer to *T. cophysa*. De Marmels and Rácenis (1982) clarify the characters and distributions of *T. calverti* and *T. cophysa*, and list only *T. cophysa* from the Galapagos. The key in De Marmels and Rácenis (1982) should be consulted. Dunkle (1989, and pers. comm.) has seen material of *T. calverti* but not *T. cophysa* from the Galapagos. My material contains only specimens of *T. calverti*. I have examined Galapagos specimens in USNM and CAS collections and found specimens of *T. calverti* which had been labelled as *T. cophysa*. If *T. cophysa* was actually once present and is now absent in the Galapagos it represents a case of natural extirpation of island populations”. Peck’s key to *Tramea* (couplet number 6) also differentiates clearly between both species:
“6a. Underside of abdomen black; abdominal segment 8 all black (Figure 6); hindwing clear with sharply edged dark basal band; male at maturity with frons all violet, male face black*Tramea cophysa*
6b. Underside of abdomen brown to red; abdominal segment 8 with a semicircular pale basal-lateral spot (Figure 5); hindwing tinged brown with an amber-edged basal band; male with only broad band on top of frons violet; female band on frons top narrower, lower frons and face otherwise pale; male at maturity with lower frons and face red*Tramea calverti*”

Our morphological analyses have shown that the *Tramea* collected in the Galapagos tally with those of *T. calverti* and not *T. cophysa;* although some differences in wing pattern and coloration between *T. calverti* and the Galapagos specimens do exist. The female lectotype of *T. darwini* (Figure 4i) possess almost no Hw basal wing spot, which likely led Kirby to describe the species as new. Variability in Hw basal wing spot pattern was also illustrated by Asahina [21] and encompasses a greater variability compared to our samples.

Regarding the male vesica spermalis, Gloger [22] stated that he was unable to find any differences in this structure between *T. darwini* and *T. cophysa* (“Ich habe die Penis aller mir zur Verfügung stehenden Exemplare des Kontinents (nur sogen. Form c, mit folgender Herkunft: Argentinien: San Isidro, Prov. Buenos Aires; Playadito, Prov. Corrientes; Mascasín, Prov. La Rioja; Bolivien: Roboré) mit denen der Galapagos-Ausbeute verglichen, ohne Differenzen zu finden. [I have compared the penis of all forms of the continent available to me (only so-called form c, with the following origin: Argentina: San Isidro, Prov. Buenos Aires; Playadito, Prov. Corrientes; Mascasín, Prov. La Rioja; Bolivia: Roboré) with those of the Galapagos yield without finding any differences.]”). At first glance, we could not find any differences in the vesica between *T. darwini* and *T. cophysa*, until we prepared the structure as described in the Materials and Methods section. Some of the various lobes present in the vesica are difficult to see, and they may be obscured in dried specimens as the ones examined by Gloger [22].

We suggest the following couplet changes in the key provided by De Marmels and Rácenis [14] for separation of *T. cophysa* from other members of the *T. cophysa* group, as follows:
“3. Head in male with dorsal surface of vertex deep metallic violet, same color as postfrons, entire postfrons metallic violet (Figure 3a); labrum mostly dark brown (Figure 3b); vesica spermalis with a small accessory lateral lobe (Figure 2c,d); underside of abdomen black in both sexes (some females with pale brown coloration but often dusted with pruinosity basally, Figure 6c), same color as lateral carina (Figure 6a,b); abdominal segment 8 all black (Figure 7a–c); hindwing clear with sharply edged dark basal band.*cophysa* Hagen
– Head in male with dorsal surface of vertex orange, basal half of postfrons metallic violet (Figure 3); labrum mostly pale brown (Figure 3d,f,h,j); vesica spermalis lacking a small accessory lateral lobe (Figure 2a,b); underside of abdomen in both sexes brown to red (Figure 6d–g); abdominal segment 8 with a semicircular pale brown basal-lateral spot (Figure 7e–h); hindwing tinged brown with an amber-edged basal band.4”

Given our results, we consider that the *Tramea* populations present in Galapagos are conspecific with *Tramea calverti*. Following the principle of priority, this species should be referred to by its older name, *T. darwini* [15]; from which *T. calverti* [37] becomes a junior subjective synonym.

All references subsequent to Kirby [15] to *Tramea* from the Galapagos Islands-Currie [18] (p. 386), Ris [17] (p. 990), Campos [19] (p. 61), Calvert [20] (pp. 228–229), Asahina [21] (p. 2), and Gloger [22] (p. 5)—listed by De Marmels and Rácenis [14] belong, rather than under *T. cophysa*, under the name *T. calverti*. The latter applied by Muttkowski as the new name for *Tramea longicauda* Brauer? var. identified by Calvert [38] (pp. 514–516) based on two males from San José del Cabo, Baja California Sur. De Marmels and Rácenis examined the surviving lectotype from Calvert [20] (p. 606).

Below, we present a new synonymy for *Tramea darwini*:


**Tramea darwini *Kirby, 1889 Status Revised***


*Tramea darwini* Kirby [15] (p. 315) (descr. ♀, Galapagos Is., Plate LI., Figure 1, entire ♀); -Currie [18] (p. 386) ♂ descr. -Kimmins [39]: 284 (lectotype designation; “The remaining four female syntypes (in bad condition) are still in the B.M. collection. This taxon is currently placed as a synonym of *Trapezostigma cophysa* (Selys)”).

*Tramea longicauda* Brauer? var.: Calvert [38] (pp. 514–516) (descr. ♂♂, Baja California, Plate xvii, Figs. 88, 89).

*Tramea cophysa?*: Calvert [16] (p. 303) (synonymy with *T. cophysa*).

*Tramea calverti* Muttkowski [37] (p. 179) (new name for *Tramea longicauda* Brauer? var.: Calvert, 1895); -Peck [13] (pp. 313, 316) (key, discussion); Dunkle [40] (p. 115) (Galapagos Islands); Gerecke et al. [6] (listed from Galapagos Islands); Peck [7] (p. 121) (“supposedly from Floreana, Santa Cruz); [the remaining synonymy given by De Marmels and Rácenis [14] (pp. 117, 118) included under this name to follow here] New Synonymy.

*Tramea cophysa* from b Ris [17] (p. 990) (descr.).

*Tramea basalis*: Carpenter [41] (p. 260) (Barbados); -Campos [19] (p. 61).

*Tramea cophysa darwini*: Asahina [21] (p. 2), Figure 1 (varibility of Hw); -Linsley & Usinger [5] (p. 126) (Isabela, Floreana, San Cristóbal, Española) -Turner [42] (pp. 288-289) (discussion variability); -Linsley [43] (p. 9) (Isabela, Floreana, San Cristóbal, Española); Peck [7] (p. 121) (distr., discussion of name for Galapagos populations).

*Tramea cophysa*: Gloger [22] (p. 5) (discussion variability) -Peck [13] (pp. 313, 316) (key, discussion);

Gerecke et al. [6] (listed from Galapagos Islands but possibly extinct).

Material Examined

*Tramea cophysa*: BRAZIL, Rio de Janeiro State: 1 ♂, Ilha de Marambaia, Praia da Armaçao (by boat); shallow exposed rain pond, 23.0425° S, 43.9517° W, 4 m, 3 December 2000, Rosser W. Garrison; São Paulo State: 1 ♂, Sergipe, Propria, ca. 21.1332° S, 50.8° W, 415 m, August, 1975; Rio Grande do Sul State: 2 ♂♂, Porto Alegre, ca. 30.0346° S, 51.2176° W, 15 m, Lema; ARGENTINA, Santa Fe Province: 1 ♀, San Cristobal, ca. 30.3167° S, 61.2333° W, 67 m), 19 February 1920, J. C. Bradley; Salta Province: 1 ♂, Quebrada de Cafayate, Hwy 68, 25.9333° S, 65.7214° W, 1500 m, 10 January 1997, Thomas W. Donnelly; 1 ♂, Dique El Tunal, pond below dam, 25.2216° S, 64.4753° W, 460 m, 27 January 2012, Natalia von Ellenrieder & Rosser W. Garrison; 1 ♀, Bosques, ca. 34.8352° S, 58.2207° W, 20 m, April 1980, A. Rodrigues Capítulo; 1 ♀, Ruta Province: 20 to Parque Nacional El Rey, 8 km from Rta. Prov. 5, pond with pleuston, 25.0167° S, 64.65° W, 790 m, 9 April 1998, Natalia von Ellenrieder; 1 ♀, pond at Nat. Rt. 50, 16 km north to Orán, 23.0082° S, 64.3667° W, 351 m, 3 November 2006, Rosser W. Garrison & Natalia von Ellenrieder; 1 ♂, 1 ♀, Arroyo Yacuy, 15 km north of Tartagal sobre ruta nacional 34, 22.371° S, 63.7725° W, 495 m, 6 November 2006, Rosser W. Garrison & Natalia von Ellenrieder; Formosa Province: 1 ♂, swamps by route 81, 12 km east of Ingeniero Guillermo N. Juarez, 23.9706° S, 61.7039° W, 0 m, 7 November 2007, Rosser W. Garrison & Natalia von Ellenrieder; 1 ♀, roadside pool on route 2, 3 km south of Mojón de Fierro, 26.0492° S, 58.0667° W, 5 November 2007; 3 ♂♂, 1 ♀, ponds 12 km south of Gran Guardia on road 16, 25.965° S, 58.9292° W, 6 November 2007; 2 ♂♂, slough 2 km S of Bañado La Estrella, 43 km north of Las Lomitas on road 28, 24.4589° S, 60.3881° W, 7 November 2007; 1 ♂, Reserva Natural Formosa, pond, 24.3167° S, 61.7978° W, 60 m, 15 February 2008, Rosser W. Garrison & Natalia von Ellenrieder; 1 ♂, Bañado La Estrella, 42 km north of Las Lomitas on road 28, 24.4589° S, 60.3881° W, 18 February 2008, Rosser W. Garrison & Natalia von Ellenrieder; 1 ♀, ponds 12 km south of Gran Guardia on road 16, 25.965° S, 58.9292° W, 6 November 2007, Rosser W. Garrison & Natalia von Ellenrieder.

*Tramea darwini*: U.S.A., Arizona: 1 ♂, Maricopa County, slough ponds by Verde River, by Ariz. Hwy. 87, Ft. McDowell Indian Reservation, 33.6758° N, 111.672° W, 1 September 1976, Rosser W. Garrison; 1 ♂, Glendale, cattle pond near Luke elementary School, ca. 33.537° N, 112.34° W, 330 m, 17 October 1961, Rosser W. Garrison; 1 ♂, Yuma County, Mohawk, Gila River at Mohawk Valley Blvd., 32.7144° N, 114.014° W, 80 m, 30 September 2002, Rosser W. Garrison & Natalia von Ellenrieder; PUERTO RICO, Manati Mun.: 1 ♂, Laguna Tortuguero, west end at Hwy 686, 18.4607° N, 66.4674° W, 2 m, 7 January 1980, Rosser W. & Jo Allyn Garrison; same data but: 1 ♂, 21 June 1981; Vega Baja Mun.: 1 ♀, Laguna Tortuguero, southeast end near Hwy 687, 18.4589° N, 66.4239° W, 4 m, 7 June 1981; Cabo Rojo Mun.: 1 ♂, Cabo Rojo, ca.18.0771° N, 67.1474° W, 30 m, 6 December 1981, Stgo. Matos; MEXICO, Veracruz State: 1 ♂, pond 20.2 km north of Alvarado, by Mex. Hwy. 180, 18.77° N, 95.76° W, 14 August 1976, Rosser W. & Jo Allyn Garrison; same data but 1 ♂, Río Coscoapan, 19.4 km east of Sontecomapan, 18.4892° N, 94.99° W, 13 m, 16 August 1976; BELIZE, Corozal District: 1 ♀, Shipstern Reserve, ca.18.3176° N, 88.1832° W, 11 m), 24 October 1992, Tineke Boomsma; VENEZUELA, Aragua State: 2 ♂♂, Lago Taguaiguai, on Cagua Rd, 10°6′59′′ N, 67°27′9′′ W, 480 m, Rosser W. & Jo Allyn Garrison; Miranda State: 1 ♀, Parcelamiento Industrial, Paracotos, ca.10°16′5′′ N, 66°56′41′′ W, 600 m; Bolívar State: 1 ♀, Canaima, palm marsh, 6°14′30′′ N, 62°50′53′′ W, 700 m, 22–25 September 1980; GUYANA, Potaro-Siparuni Region: 1 ♀, Konawaruk watershed, Mango Landing, Essequibo River, 5.315° N, 58.9067° W, 41 m, 19 September 2014, Rosser W. Garrison & Richard Mohabie; FRENCH GUIANA, 1 ♀, Piste de Kaw, about PK 18, ca.4.6202° N, 52.2889° W, 250 m, 17 February 1988, Rosser W. Garrison; same data but: 1 ♂, 3 ♀♀, small canal 17 km south of Tonate on route D5, 4.8708° N, 52.5192° W, 18 m, 18 February 1988; same data but: 1 ♂, Approuague-Kaw, Kaw Mountain, 311 mao; light trap (FRG TRAP XL 2), 4.566° N, 52.2053° W, 325 m, 14 February 2007, N. Jönsson; ECUADOR, Galápagos: 1 ♂, 1 ♀, Galápagos, Isla Santa Cruz, Cerro Mesa; ACR-05182, ACR-05183, 0.6433° S, 90.2876° W, 425 m, 28 February 2014, Adolfo Cordero-Rivera; 2 ♂♂, Isla San Cristobal, nearby the Cucuve Eco Hostal, 0.91° S, 89.589° W, 97 m, 1 August 2018, M. Olalla Lorenzo-Carballa; same data but: 2 ♀♀; 1 ♂, 1 ♀, Galápagos, Isla Isabela, El Chapin, 0.9452° S, 90.9743° W, 11 m, 15 August 2018;same data but: 1 ♂, 1 ♀, Isla San Cristobal, Finca Guadalupe, 0.9266° S, 89.4862° W, 190 m, 2 August 2018; BRAZIL, Rondônia State: 2 ♂♂, Fazenda Rancho Grande, 62 km southwest of Ariquemes, 10.53° S, 62.8° W, 165 m, 2–11 November 1989, Rosser W. Garrison; Rio de Janeiro State: 2 ♂♂, 1 ♀, Ilha de Marambaia, Praia da Armacao (by boat), 23.0425° S, 43.9517° W, 4 m, 3 December 2000, Rosser W. Garrison; ARGENTINA, Salta Province: 1 ♂, Arroyo Yacuy, 15 km north of Tartagal sobre ruta nacional 34, 22.371° S, 63.7725° W, 495 m, 6 November 2006, Rosser W. Garrison & Natalia von Ellenrieder; Formosa Province: 1 ♀, swamps by route 81, 12 km east of Ingeniero Guillermo N. Juarez, 23.9706° S, 61.7039° W, 0 m, 7 November 2007, Rosser W. Garrison & Natalia von Ellenrieder.

## 5. Conclusions

Our genetic analyses show that the *Tramea* species from Galapagos belongs to a clade that comprises also the continental species *T. calverti*, and *T. cophysa* appears as a closely related species.Concordant with the results of the genetic analyses, the morphology of the *Tramea* collected in Galapagos is closer to *T. calverti* than to *T. cophysa*.Only one morphological character has been found which consistently discriminates between both species: an accessory lobe in the male sperm vesicle, which appears in *T. cophysa*, but is absent in both *T. calverti* and the *Tramea* from Galapagos.Given these results, and following the taxonomic principle of priority, the *Tramea* species currently found in Galapagos should be referred to by its older name, *T. darwini*.

## Figures and Tables

**Figure 1 insects-12-00021-f001:**
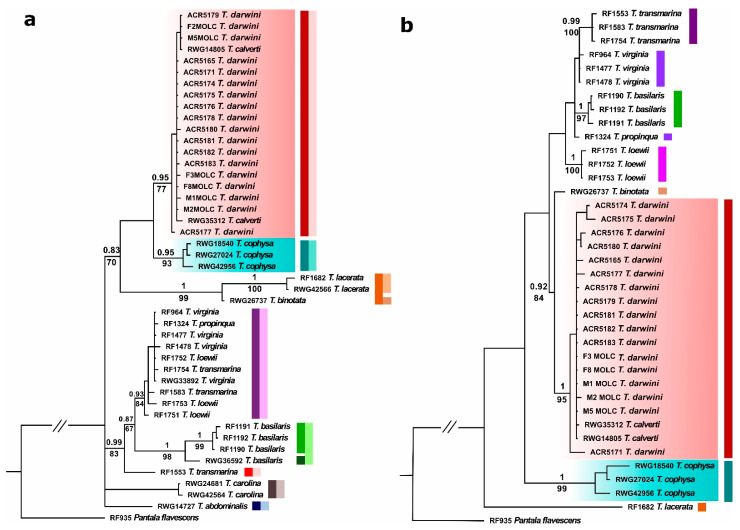
Tree representing the phylogenetic relationships among the *Tramea* species analyzed in this study, using (**a**) mtDNA (16S + COI) and (**b**) nDNA (ITS) data. Numbers above and below branches represent Bayesian posterior probability values and ML bootstrap values, respectively. The clade comprising the *Tramea* from Galapagos and *T. calverti* is highlighted in pink, and the *T. cophysa* clade appears highlighted in blue. The colored bars by each clade represent the proposed species delimitation based on Automated Barcode Gap Detection (ABGD) analyses, where each color represents a delimited species. For the mtDNA markers, the dark and light colour bars represent species delimitations according to COI and 16S datasets, respectively.

**Figure 2 insects-12-00021-f002:**
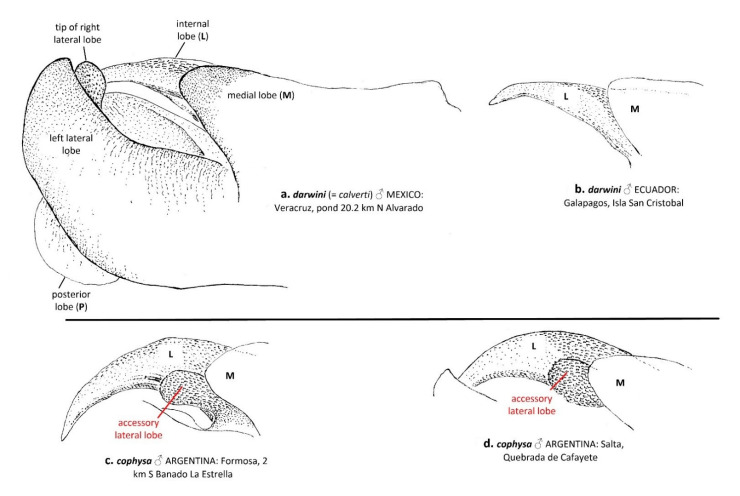
Lateral view of the male vesica spermalis of *T. darwini* (senior synonym of *T. calverti*, stat. rev.; **a**,**b**) and *T. cophysa* (**c**,**d**), showing the accessory lateral lobe present in the latter, a characteristic that allows for discrimination between both species. Note that the left lateral lobe, the posterior lobe and the tip of the right lateral lobe appear only illustrated in a.

**Figure 3 insects-12-00021-f003:**
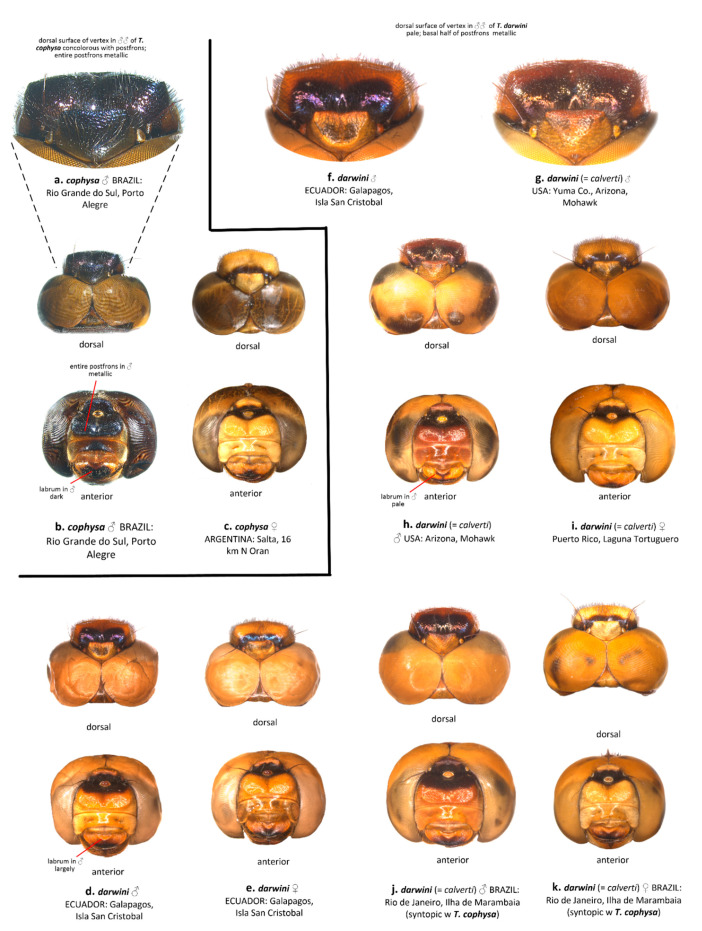
Dorsal and anterior view of the head of males and females of *T. cophysa* (**a**–**c**) and *T. darwini* (senior synonym of *T. calverti*, stat. rev.) (**d**–**k**), showing the differences in coloration of vertex, postfrons and labrum.

**Figure 4 insects-12-00021-f004:**
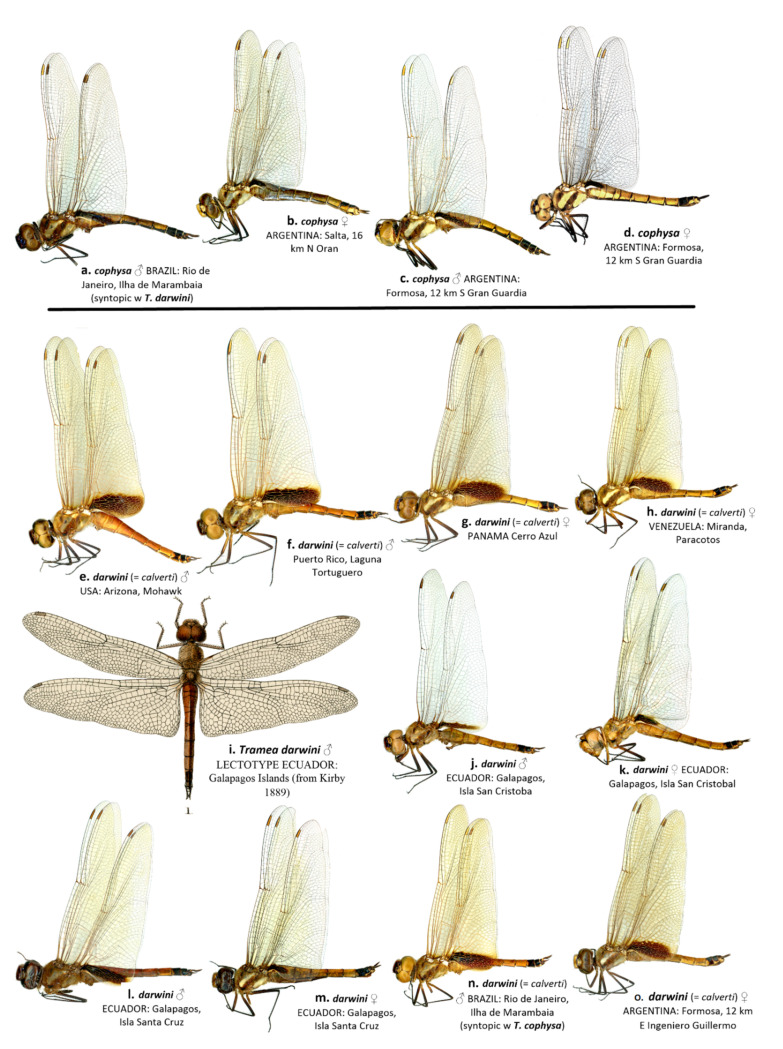
Lateral view of habitus of *T. cophysa* (**a**–**d**) and *T. darwini* (senior synonym of *T. calverti*, stat. rev.) (**e**–**o**). Dorsal view of the lectotype of *T. darwini* Kirby 1889 is also included (**i**).

**Figure 5 insects-12-00021-f005:**
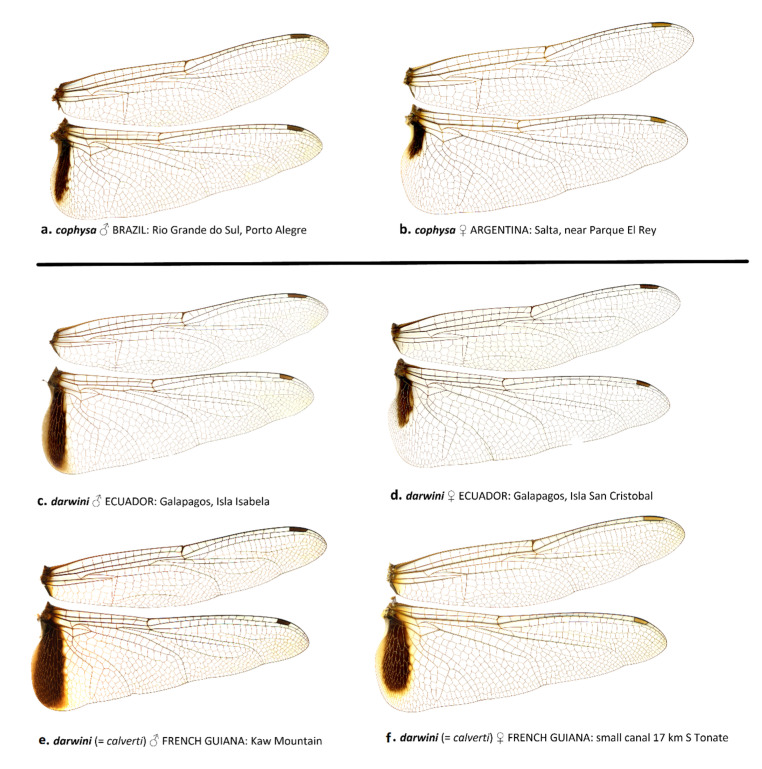
Wing scans of *T. cophysa* (**a**,**b**) and *T. darwini* (senior synonym of *T. calverti*, stat. rev.) (**c**–**f**).

**Figure 6 insects-12-00021-f006:**
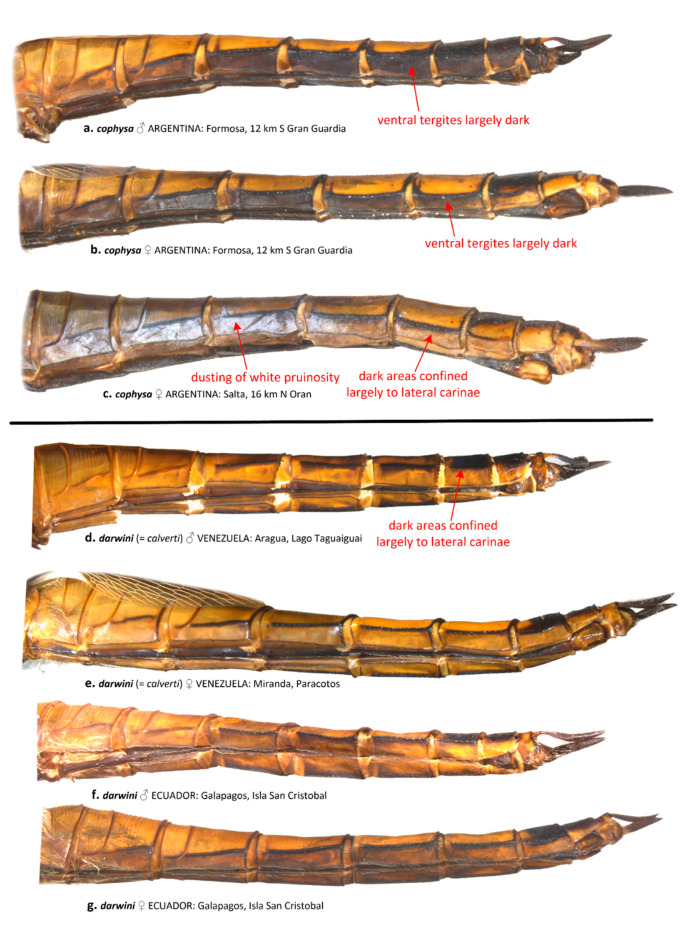
Left ventrolateral view of the abdomen of *T. cophysa* (**a**–**c**) and *T. darwini* (senior synonym of *T. calverti*, stat. rev.) (**d**–**g**).

**Figure 7 insects-12-00021-f007:**
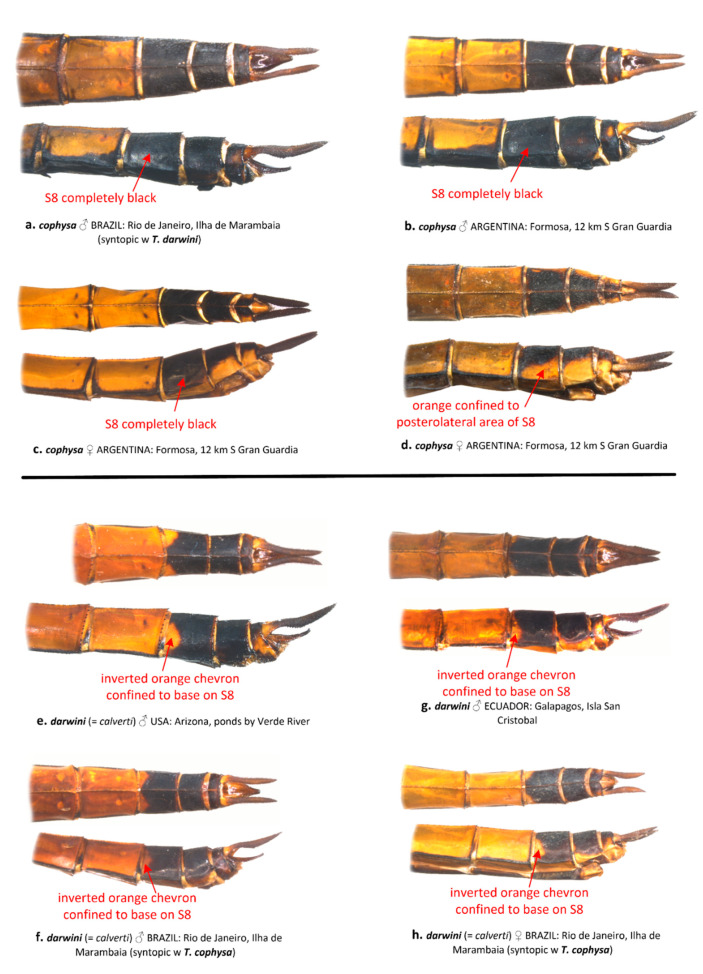
Dorsal and lateral views of abdominal segments S7-10 from *T. cophysa* (**a**–**d**) and *T. darwini* (senior synonym of *T. calverti*, stat. rev.) (**e**–**h**).

## Data Availability

DNA sequences obtained from this study are available in the GenBank database, and data are available in the Supplementary Information File. All the *Tramea* specimens examined and/or used for genetic analyses are deposited in the collections of Adolfo Cordero-Rivera and Rosser W. Garrison.

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
