# Peer review of "Darwin Returns to the Galapagos: Genetic and Morphological Analyses Confirm the Presence of Tramea darwini at the Archipelago (Odonata, Libellulidae)"

_insects, 2020, doi:10.3390/insects12010021_

Round 1

Reviewer 1 Report

see following comments "Review of Tramea darwini ms."

REVIEW OF:

Darwin returns to the Galapagos: genetic and morphological analyses confirm the presence of Tramea darwini at the archipelago (Odonata, Libellulidae), by M. Olalla Lorenzo-Carballa, Rosser W. Garrison, Andrea C. Encalada & AdolfoCordero-Rivera

Simple Summary: T. darwini has a range larger than what I would call "some parts of continental America," being very widespread from far southern US well down through South America, as well as widely through the Caribbean.

Line 22 - Tramea is misspelled

Abstract:  Lines 36-37 - Shouldn't it be that the accessory lobe is present in T. cophysa but lacking in T. calverti and the Tramea from the Galapagos?

As the purpose of the paper is to question the taxonomic status of darwini, I would list it throughout as "Tramea (or T.) darwini" rather than "Tramea cophysa (or T. c.) darwini." I think doing it the latter way is quite confusing. And in fact, you definitely shouldn't list it that way in your Fig. 1, as putting calverti and darwini together in the same clade makes it look too much like "T. c. darwini" is short for Tramea calverti darwini. I was confused when I first looked at the figure, as of course "c" also could stand for calverti. I really think that the "c" should be eliminated everywhere it has been used.

I personally would have called calverti by name throughout the paper rather than labeling figures "darwini (= calverti)". Again, I think that makes reading the paper more confusing.

Then rather than ever having to say "the Tramea from the Galapagos," it could be made clear in the introduction that the name darwini always referred to the Tramea from the Galapagos.

In Supplementary Table S1, seeing T. darwini (= calverti) is again quite confusing, as the paper concludes in the end that T. calverti = T. darwini.

100 - "among" is preferred to "between" when comparing more than two

Figure 1. The writing is awfully small, even enlarged on my computer to a size considerably larger than the print edition will be. Some species names were almost illegible. I strongly recommend enlarging them or abbreviating them to make them more legible to those readers who will be very interested in the results.

As far as I know, Tramea propinqua is a synonym of T. transmarina. I hope that in molecular studies you can change the name of a sample from GenBank to bring it up to present-day taxonomy.

It doesn't seem to me that mtDNA shows relationships among other Tramea species all that well. nDNA clearly does a better job.

In Fig. 2, it wouldn't be a bad idea to put an "L" right on the lateral lobe, so the reader can quickly see the comparison. Again, the writing will be very small in the published paper. And the lobes of 2b actually seem to be part of the 2a structure. 2b or not 2b? That is the question.

339 - I presume this should be "form" rather than "from"

342 - this presumably should be "discussion of variability"; ditto line 345

I did not examine either the Material Examined or the References in the detail needed to see if there were any small errors there. However, I did notice that on line 520, the Linsley reference didn't get a number, and that may have messed up the numbering of the subsequent references. So out of curiosity, I looked for those references, and I found no reference to either Dikstra et al. or Palumbi, nor the numbers 40 and 41. I did find Linsley, listed as 39. Perhaps a further check of the references is in order.

409 - I wonder if "genetic analysis" might be a better term than "molecular analyses;" I didn't look for this elsewhere

Reviewer 2 Report

The manuscript entitled “Darwin returns to the Galapagos: genetic and morphological analyses confirm the presence of Tramea darwini at the archipelago (Odonata, Libellulidae)” is a valuable work. The manuscript is well written and linear. The introduction provides an exhaustive background for understanding the study, the main aim of the study is clear and the performed analyses in line with it. Discussion and conclusions are relevant with the results obtained. For these reasons the manuscript deserves to be published, below few minor comments and suggestions for improving the present version of the manuscript.

Introduction

In order to standardize I suggest mentioning T. darwini always as T. cophysa darwini in the introduction section since it was the valid name of this taxon before your work.

Material and Methods

Line 96: it is not clear to me if “RWG’s personal collection” is a dry collection. It is important since DNA extractions from dry preserved specimens are frequently problematic for the risk of contamination. I suggest to clarify this point. In addition, if you extracted DNA from specimens conserved in a dry collection, have you checked the obtain sequences to ensure they are not derived from contamination?

Line 102: I suggest to include adopter primers citations also in the Material and Methods section, not only in the supplementary table.

In order to definitively confirm the delimitation of the species of the T. cophysa group basing on molecular information, I suggest to perform molecular species delimitation analyses through molecular species delimitation methods. Since the data you produce allow it, you could perform single locus species delimitation using distance-based methods (e.g. ABGD, Puillandre et al., 2012) or tree-based methods (e.g. PTP, bPTP, mPTP not requiring ultrametric trees, Zhang, Kapli, Pavlidis, & Stamatakis, 2013; Kapli et al., 2017) or both, but you could also perform a multilocus species delimitation using methods such as BBP (Yang, 2015; Fluori et al., 2020). Such kind of analyses could be a nice addition to the work, which is per se, also in the actual form, a really good contribution to Odonata taxonomy.

Results

Line 178-181: I suggest to report pairwise nucleotide distances for 16s and COI separately, since the substitution rate of these genes could be different and also the 2% threshold between intra/interspecific distances you mention in the discussion is estimated on COI not on both genes.

Reviewer 3 Report

The authors present compelling molecular and morphological data supporting their conclusion that the Galapagos specimens they examined belong to Traema calverti, and not to Traema cophysa. The phylogenetic trees as well as the genetic pairwise distances both provide strong support. However, it was very nice to see distinct morphological evidence as well (in addition to the explanation as to why this feature may have been missed in previous studies). The images and descriptions of all of the morphological features that were compared across the specimens were very thorough and well presented. It was also very useful that the authors recommended specific changes that serve to update the key by De Marmels and Rácenis. Overall this is a sound and thorough contribution to the literature and in the opinion of this reviewer, ready for publication with no recommended changes. 
